# Mucosal Immunity to Gut Fungi in Health and Inflammatory Bowel Disease

**DOI:** 10.3390/jof9111105

**Published:** 2023-11-14

**Authors:** Sean L. Carlson, Liya Mathew, Michael Savage, Klaartje Kok, James O. Lindsay, Carol A. Munro, Neil E. McCarthy

**Affiliations:** 1Centre for Immunobiology, The Blizard Institute, Queen Mary University of London, London E1 2AT, UK; 2Gastroenterology Department, Royal London Hospital, Barts Health NHS Trust, London E1 1BB, UK; 3Aberdeen Fungal Group, Institute of Medical Sciences, University of Aberdeen, Aberdeen AB24 3FX, UK

**Keywords:** inflammatory bowel disease, Crohn’s disease, ulcerative colitis, *Candida albicans*, gut microbiome

## Abstract

The gut microbiome is a diverse microbial community composed of bacteria, viruses, and fungi that plays a major role in human health and disease. Dysregulation of these gut organisms in a genetically susceptible host is fundamental to the pathogenesis of inflammatory bowel disease (IBD). While bacterial dysbiosis has been a predominant focus of research for many years, there is growing recognition that fungal interactions with the host immune system are an important driver of gut inflammation. *Candida albicans* is likely the most studied fungus in the context of IBD, being a near universal gut commensal in humans and also a major barrier-invasive pathogen. There is emerging evidence that intra-strain variation in *C. albicans* virulence factors exerts a critical influence on IBD pathophysiology. In this review, we describe the immunological impacts of variations in *C. lbicans* colonisation, morphology, genetics, and proteomics in IBD, as well as the clinical and therapeutic implications.

## 1. Introduction

Inflammatory bowel disease (IBD) is a group of chronic inflammatory conditions affecting the intestine that are subclassified into Crohn’s disease (CD) and ulcerative colitis (UC). CD causes patchy transmural inflammation that can affect any part of the gastrointestinal tract, whereas UC is defined by continuous mucosal inflammation of the colon. IBD pathogenesis is thought to arise from dysregulation of the immune system in genetically predisposed individuals exposed to environmental triggers. There is a rising global burden from chronic inflammatory and autoimmune diseases, including IBD, which is increasing too quickly to be due to genetics alone [1,2]. The human immune system is highly variable between individuals, predominantly due to non-genomic influences such as microbial communities in different organ systems and environmental factors, which are potential contributors to this increasing prevalence [2].

The gastrointestinal tract is a sophisticated immune organ that has evolved to maintain a diverse microbial community through a range of tolerance mechanisms whilst restricting the growth of pathogens. We now appreciate that the microbiota plays a major role in several symbiotic relationships with the human host. Metagenomic data have demonstrated how a diverse microbiome is important to sustain the mucosal barrier integrity, immune tolerance of commensal organisms, and host metabolism [3]. Host–microbiota crosstalk can promote resistance to enteric pathogens, limit gut barrier injury, and thus prevent disseminated infection [4,5].

Our understanding of the bacterial microbiome in human health and disease has increased significantly in recent years, with evidence of dysbiosis being reported in a wide range of disorders. Intestinal dysbiosis can broadly be defined as an alteration in the gut microbial community that functionally contributes to intestinal or systemic pathology [6]. The key features of bacterial dysbiosis in IBD patients are a reduced diversity, lower levels of ‘good’ bacteria such as Firmicutes, and an increase in potentially pathogenic bacteria [7,8,9,10]. Far less is known about other key organisms within the gut ecosystem. Historically, identification methods for intestinal bacteria have been significantly more advanced than for other organisms, with large metagenomic studies as well as novel culture methods enabling the characterisation, quantification, and culturing of bacteria [11,12]. Fungal identification methods are not as advanced, and there are a lack of metagenomic databases to support the cross-referencing of the multiple species that could potentially influence IBD pathogenicity. A further challenge is the major differences between human and rodent mycobiomes. Although *Candida* spp. are now known to colonise the gut in most humans, they are not natural commensals in the rodent intestine [13]. Among the gut fungi, *Candida albicans* appears to play a pivotal role in shaping human host immunity, a concept that was first identified in IBD through research into anti-*Saccharomyces cerevisiae* antibodies (ASCA) and anti-glycan antibodies.

## 2. ASCA and Other Anti-Glycan Antibodies

An association between intestinal fungi and IBD has long been suspected, given the early research linking ASCA and CD. ASCA is an anti-glycan antibody against mannan, a conserved component of fungal cell walls, that serves as a serological biomarker for disease onset and clinical course in IBD [14]. *C. albicans* and other yeasts can induce the production of ASCA as well as other anti-glycan antibodies associated with CD, namely anti-mannobioside (AMCA), anti-laminaribioside (ALCA), and anti-chitobioside (ACCA), although their clinical use is inferior to that of ASCA [14,15,16,17].

The ASCA status varies depending on the IBD type and disease location. Approximately 50 to 60% of CD patients are ASCA positive, compared to only 10–15% of UC patients and 5% of the general population. Patients with small bowel Crohn’s disease also exhibit higher rates of ASCA positivity compared to those with isolated colonic disease [18,19]. ASCA levels are associated with the NOD1 and NOD2 genetic variants, polymorphisms that predispose an individual to the development of IBD [20]. ASCA has good specificity for the diagnosis of CD but poor sensitivity, so it can be used in conjunction with anti-neutrophil cytoplasmic antibodies (pANCA) to differentiate between CD and UC in the clinic [18,21,22,23]. Detectable ASCA levels can pre-date the development of IBD by several years but they are a poor prognostic marker of CD; adult studies have shown that ASCA is associated with an increased risk of stricturing and penetrating disease as well as a need for early surgery [24,25,26,27]. In paediatric cohorts, however, reports on the association between ASCA and disease severity are conflicting [28,29,30].

Research into the associations among *C. albicans*, the ASCA status, and CD has produced mixed results. Although *C. albicans* is an immunogen for ASCA, a study into familial CD found that ASCA levels were stable and high in CD patients independent of colonisation, despite *C. albicans* being more likely to colonise patients with CD and their first-degree relatives [14,31]. In a separate study, in CD patients, *Candida tropicalis* was the only fungus that positively correlated with the ASCA level. This study, however, reported a surprisingly low *C. albicans* abundance in both CD and healthy controls compared to other similar studies [9]. Lastly, a recent Australian study of a paediatric CD cohort identified that the bacterial microbiome varies according to the ASCA status, although the specific role of *C. albicans* was not investigated [22]. While current European IBD guidelines do not recommend the use of ASCA and pANCA for differentiating UC and CD given the poor sensitivity, this research did draw attention to the role of fungi in IBD pathogenesis [32].

## 3. The Mycobiome

Much like the bacterial microbiome, the mycobiome is imbalanced in IBD with differing patterns reported in CD and UC (Figure 1). There is mounting evidence that fungal diversity and the global fungal load are increased in CD, especially in inflamed tissue where there is an expansion of the *Candida* spp. diversity that correlates with disease activity [8,33]. However, Sokol and colleagues found reduced fungal diversity in UC and colonic CD, whereas the microbiome in ileal CD favoured fungi over bacteria [34]. These findings were corroborated by other studies, which showed higher fungal richness in CD compared to UC and healthy controls [35]. Moreover, there was a strong correlation between reduced fungal and bacterial diversity in UC, suggesting inter-kingdom interactions that are not seen in CD [34]. In contrast to adult studies, a paediatric study found decreased fungal diversity in both CD and UC, but with similar discrete fungal communities. In particular, there was a higher abundance of *Candida* spp. [7].

There are distinct mycobiome shifts in IBD compared to the general population. Numerous studies have observed an increased proportion of *C. albicans* and a decrease in *S. cerevisiae*, with these variations being amplified in patients with active disease [8,34,36,37]. A more recent study evaluating the mycobiome in UC patients reported the same findings for *C. albicans*; however, the authors also detected a marked expansion of *Saccharomyces* species [38]. The geographical location has a significant impact on these shifts, with Japanese and Western cohorts displaying varying patterns in the mycobiome diversity. These data suggest a potential influence of regional mycobiome profiles on the IBD phenotype and severity, as well as variable therapeutic efficacy in diverse populations. A constant feature, however, is the increased presence of *C. albicans* in disease [39].

Inter- and intra-kingdom interactions that are not yet fully understood occur within the gut. *Faecalibacterium prausnitzii* was shown to inhibit the growth of *C. albicans*, virulence factor expression, and transition to pathogenic behaviour in vitro [40]. In a dextran sulphate sodium (DSS) colitis murine model, mice co-infected with *F. prausnitzii* and *C. albicans* had less severe colitis with a significantly reduced mucosal Candida burden compared to mice infected with *C. albicans* alone [40]. A proposed mechanism is the induction of NLRP6 inflammasome activity with elevated secretion of IL-1β, IL-18, and antimicrobial peptides [40]. A separate study reported that *Escherichia coli* can inhibit *Candida* spp. biofilm and hyphal development [41]. A recent pre-print article identified that the ubiquitous UC treatment 5-aminosalicylic acid can reduce post-antibiotic *C. albicans* bloom, which is thought to be promoted by the restoration of mucosal hypoxia and gut re-colonisation by Clostridia [42]. Separately, *Candida tropicalis* levels appear to be positively correlated with several pathogenic bacteria within the guts of CD patients, and their co-existence within biofilms is associated with more filamentous growth [9]. The interplay between different organisms within the microbiome is an important ongoing field of research with potential implications for IBD.

## 4. *Candida albicans* Pathobiology

There are several *Candida* species, among which *C. albicans* is the most prevalent human pathogen. This polymorphic fungus is a commensal coloniser of the human skin and mucosal surfaces where it can exist in yeast and filamentous hyphal forms. Previous reports suggest that the human GI tract is colonised by *Candida* species in up to 60% of healthy individuals, typically in the yeast form that establishes symbiotic commensalism with the host [43,44], although more recent studies using targeted metagenomics have reported that gut colonisation is essentially universal in healthy adults [13]. There is significant genetic diversity within *C. albicans* gut isolates which impacts the colonising potential of different strains. Certain transcriptional regulators, such as Efg1, control commensalism by repression of the hyphal growth gene *EFH1* [45]. *C. albicans* hyphae have pathogenic potential through the invasion of the epithelial cell layer and the release of cytolytic enzymes [46,47]. Hyphal growth also provides a mechanism of immune evasion by facilitating escape from phagocytes after engulfment [48]. Temperature, pH, O_2_/CO_2_ tension, and interactions with other gut organisms all affect yeast-to-hyphal morphological switching, and understanding how these environmental factors induce *C. albicans* pathogenicity could be key when considering possible therapeutic targets [49].

There is a distinction between fungi that exist in close proximity to the intestinal mucosa, typically immunogenic species such as *Candida* and *Saccharomyces* species, and luminal fungi that tend to be transient in nature [50]. Host–fungal symbiosis with mucosa-associated *C. albicans* can lead to the promotion of the gut barrier integrity through JAK/STAT signalling, Th17 cell responses, and IL22 secretion [50]. *C. albicans*, however, is a well-recognised opportunistic pathogen in immunocompromised and intensive care patients, with the gut microbiome being the main reservoir for infection and dissemination [51]. Further to this, IBD patients on immunosuppressive medications, such as steroids, are at an especially high risk of developing invasive fungal infections [52]. Given the significant morbidity caused by this organism, *C. albicans* was recently included on the WHO fungal priority pathogen lists [53], emphasising the clinical importance of this microbe and the need for further research into its mechanisms of virulence.

## 5. Key Virulence Factors: Hyphae, Adhesins, and Invasins

*C. albicans* expresses several major virulence factors, with adhesion to hosts’ epithelial surfaces being a critical first step in establishing infection. Adhesion factors are crucial for enabling *C. albicans* to bind to and remain associated with the gut epithelium [54]. Adhesins exist on the outer cell wall, and include the *ALS*, *HWP*, and *IFF/HYR* gene families. Members of the agglutinin-like sequence (Als) family act as both epithelial adhesins and invasins. Als3, in particular, is a hyphae-specific surface protein that binds to host epithelial cell receptors such as E- and N-cadherin, thereby triggering endocytosis [55]. Inhibition of *C. albicans* adhesion to epithelial surfaces is conceptually an attractive therapeutic target, and several novel small molecules and monoclonal antibody drugs have shown promising results [56]. Als3 is especially interesting, given that a vaccine targeting this protein has been developed and trialled in vaginal candidiasis, a topic we explore later in the review.

A central factor in both the development of epithelial damage and immune evasion is the yeast-to-hyphal transition. Several transcription factors (e.g., Efg1 and Ume6) control the yeast–hyphae transformation [57]. The expression of these factors can affect the ability of *C. albicans* to exist as a commensal organism. For example, Ume6 is a filamentation regulator that inhibits commensalism via the downstream activation of proteases and adhesion factors [58]. Depending on the anatomical site, hyphae have been shown to exert varying effects. In a murine model of oral candidiasis, the morphology and hyphal growth of *C. albicans* did not correlate with the degree of histological inflammation [59]. In contrast, DSS colitis is more severe in mice colonised with filamentous *C. albicans* compared to the yeast form, and Als adhesins were shown to be crucial in orchestrating this [60]. An in vitro analysis by Li and colleagues supported this scenario, with the cellular toxicity of *C. albicans* being linked to filamentation [36]. They expanded on these findings by ablating enhanced filamentous growth protein 1 (Efg1) in gut-derived *C. albicans* isolates, which blunted the ability of ‘high damage’ filamenting strains to inflict host cell injury [36]. Importantly, variation in the filamentation ability, even within *C. albicans* strains from the same host, was independent of the disease status [36]. These results demonstrate the importance of hyphae as virulence factors, but also highlight that the variable pathogenicity also depends on the anatomical site and other virulence factors.

In *C. albicans*, the *ECE1* gene encodes a novel cytolytic peptide known as candidalysin that mediates epithelial cell damage [47]. Candidalysin expression requires specific proteinases to mediate the post-translational conversion of immature to mature toxins, at which point the active peptide is released from the hyphae into epithelial ‘invasion pockets’ [61]. Epithelial cells are induced to release alarmins that activate the EGFR and MAPK pathways [47,62,63]. The subsequent antifungal inflammatory cascades lead to both IL-1 and IL-36 induction and the amplification of IL-17 expression [64,65]. The combination of epithelial adhesion, hyphal growth, and candidalysin secretion into invasion pockets triggers mucosal inflammation and epithelial damage through oxidative stress and the activation of necrotic, rather than apoptotic, pathways [66,67]. In addition to this role in cellular damage, candidalysin expression is also thought to constitute a mechanism of immune evasion, since this toxin can activate the NLRP3 inflammasome to trigger mononuclear phagocyte cell death after fungal engulfment [68]. Interestingly, differences in *ECE1* gene expression between *C. albicans* strains were not found to influence keratinocyte damage levels in an in vitro model, although there was a strong correlation between cell injury and IL-1α release in these same cultures [59]. Conversely, a variant *ECE1* allele identified in a vaginal *C. albicans* strain was associated with reduced secretion of the candidalysin peptide, leading to diminished virulence [69]. In addition to intra-strain variation, different gut-resident *Candida* spp., such as *C. dubliniensis* and *C. tropicalis*, produce candidalysin of variable potency levels to induce cellular damage and cytokine responses [70]. Intra-strain variation in virulence factors is increasingly being recognised as one of the most important factors determining *C. albicans* pathogenicity.

## 6. *C. albicans* Intra-Species Variation

*C. albicans* intra-species diversity is a critical determinant of pathogenicity in human hosts. There is significant genetic diversity in *C. albicans* gut isolates between IBD patients that is independent of the disease status and geographic location [36]. Genomic variations lead to alterations in the cellular phenotype, hyphae formation, mycotoxin production, and overall virulence [36,71,72]. Using a murine oropharyngeal candidiasis (OPC) model, Schonherr and colleagues reported differences in host persistence and the inflammatory response between specific *C. albicans* isolates, with these outcomes being inversely correlated [59]. Proinflammatory strains of *C. albicans* triggered an increased neutrophil response and IL-17 pathway activation through alarmin release, resulting in significant mucosal inflammation that, in turn, led to rapid fungal clearance. Interestingly, the filamentation ability did not significantly vary between the pro- and anti-inflammatory strains, indicating an alternative cause for the differences in host persistence and cell damage [59].

In a recently published study, *C. albicans* strains were categorised into low- and high-damage groups based on their ability to injure murine macrophages, referenced against a known highly virulent strain, SC5314 [36]. High-damage (HD) strains induced a significantly stronger Th17 cell inflammatory response in colonised mice compared to low-damage strains (LD), contributing to intestinal inflammation in a DSS-induced colitis model [36]. These results were independent of the fungal load, as each strain colonised mice to similar levels [36]. Following this theme, in ulcerative colitis, the functional characteristics of different gut *C. albicans* strains, rather than the relative abundance, resulted in candidalysin-dependent macrophage IL-1β production which, in turn, correlated clinically with endoscopic disease severity, as assessed by Mayo scores [36]. Interestingly the impact of *C. albicans* strain variation has also been studied in other gut disorders, including irritable bowel syndrome, where the importance of the genetic and phenotypic diversity was highlighted at the strain level [73]. The mucosal immune system has developed measures to target *C. albicans* variants with high expression of virulence factors in an attempt to prevent tissue invasion and barrier damage (Figure 2).

## 7. Mucosal Immunology

Secretory IgA (sIgA) is an antibody involved in intestinal immunity and homeostasis that coats microbes and controls the host immune response to commensal and pathogenic bacteria [74]. It is secreted by plasma cells in the colonic mucosa and Peyer’s patches of the small intestine, induced by food antigens and intestinal micro-organisms [74]. In the large intestine, antifungal IgA is predominantly induced by filamentous *C. albicans*, with sIgA preferentially targeting hyphal formation as well as adhesins and candidalysin [60,75]. sIgA manipulates fungal virulence factors by reducing the expression of hyphae-produced proteins, such as Ece1-derived candidalysin, Als adhesins, and Sap proteinases, which, in turn, improves *C. albicans* commensalism [60]. Interestingly, CD patients have a decreased antifungal IgA response to hyphal virulence factors, which could feasibly contribute to the *C. albicans* pathogenicity in IBD [75]. sIgA is one component of a sophisticated antifungal immune response induced by *C. albicans*.

## 8. C-Type Lectin Receptors and CARD9

Antifungal immunity is dependent on signalling cascades triggered by pattern recognition receptors (PRRs) that sense fungal pathogen-associated molecular patterns (PAMPs) such as β-glucans. C-lectin receptors (CLRs) are a key family of PRRs that mediate antifungal signalling through caspase recruitment domain 9 (CARD9)-dependent pathways [76]. GWAS studies have implicated polymorphisms in both CARD9 and C-lectin receptor genes with IBD and fungal disease [77,78].

Dectin-1 was the first C-lectin receptor to be identified and was initially described in myeloid lineage cells. High expression of Dectin-1 in monocytes, macrophages, dendritic cells, and neutrophils facilitates the innate detection of β-glucan ligands, leading to the rapid production of reactive oxygen species and the upregulation of cytokines including IL-1β, IL-6, IL-23, and TNF-α [79]. Macrophage sensing of *C. albicans* yeasts is dependent on Dectin-1, which results in phagocytosis by binding β-1,3-glucans and triggering Toll-like receptor (TLR) pathways [80]. CLRs are also involved in adaptive immunity and have been associated with the differentiation of CD4+ T cells to Th1 and Th17 cell fates [79]. Dectin-1 mediates antifungal responses by triggering a CARD9-dependent inflammatory cascade that signals through MAPK and NF-κB to drive the expression of IL-17A, IL-22, and IL-1β [81,82,83,84]. The relevance of this pathway in IBD is underscored by the fact that mice deficient in Dectin-1 are more susceptible to DSS colitis compared to wild-type controls, with elevated IFNγ, TNFα, and IL-17, as well as an increased histological severity [78]. In the same study, the Dectin-1-deficient mice were also prone to pathogenic fungal infections, accompanied by an increased proportion of tissue-invasive *C. tropicalis* and a decrease in non-pathogenic *Saccharomyces* spp. in their microbiome. Interestingly, concomitant fluconazole treatment reduced the colitis severity [78], whereas human variants in the Dectin-1 gene CLEC7A have been associated with a severe UC phenotype [78]. The fact that variations in Dectin-1 function severely impact both the antifungal immunity and IBD suggests that there are strong links between these observations.

The caspase recruitment domain 9 (CARD9) gene encodes an adaptor protein that mediates PRR signalling and the downstream induction of antifungal immunity [85]. Different CARD9 variants can either be protective or deleterious in IBD, and importantly, autosomal recessive loss-of-function mutations are a recognised cause of life-threatening fungal infections [86,87,88,89]. CARD9 deficiency in murine colitis models has shown interesting, yet somewhat conflicting, results. In one study, CARD9-deficient mice displayed reduced Th1 and Th17 cytokine responses to DSS colitis with an increased intestinal fungal burden [90]. A subsequent study suggested that a lack of tryptophan metabolism and IL-22 downregulation were responsible for alterations in bacterial and fungal microbiota in CARD9-deficient mice, while IL-17 levels were unaffected [91]. Most recently, Danne et al., reported that the expression of CARD9 in neutrophils, and not lymphocytes or epithelial cells, protected against murine colitis [92]. CARD9 deficiency in neutrophils led to premature cell death and the release of reactive oxygen species, and importantly, reduced the fungicidal ability [92]. The association between altered CLR and CARD9 function and IBD pathogenesis through defective fungal immunity is clearly very complex, and further research into the relative contributions of different immune cells will be crucial.

## 9. Myeloid Cells

Neutrophils instigate and orchestrate mucosal defence against microbes, but can also induce pathological inflammation through excessive recruitment and activation [93]. An important consideration when using laboratory mice as a model organism is the paucity of granulocytes relative to ‘wild’ rodents and humans. Exposing laboratory mice to environmental fungi, including *C. albicans*, induces granulopoiesis through IL-6 and candidalysin signalling, resulting in the sustained expansion of neutrophil populations [94]. Several chemokine-dependent mechanisms of neutrophil mucosal recruitment have been identified in IBD. IL-8, GM-CSF, and IFN- γ, as well as fungal proteins such as candidalysin, exert important influences on the process [95,96,97]. The major role of IL-1 as a neutrophil chemokine in antifungal immunity was demonstrated in a recent paper by Gander-Bui et al. Macrophage-secreted IL-1 receptor antagonist (IL-1Ra) blocked neutrophil recruitment in invasive candidiasis, thereby increasing murine mortality, whereas the therapeutic neutralisation of IL-1Ra reversed these harmful effects [98]. Viral-induced type I interferons were shown to promote macrophage secretion of IL-1Ra [98].

Once resident in mucosal tissue, neutrophils contribute to intestinal inflammation through several mechanisms, including antifungal immune responses. Neutrophil extracellular traps (NETs) are antimicrobial structures composed of chromatin and microbicidal molecules such as cathelicidin and calprotectin, peptides that are both associated with IBD [99,100]. The formation of NETs is the result of a unique form of neutrophil cell death termed ‘NETosis’ which can be induced by both yeast and filamentous forms of *C. albicans*. Neutrophils are capable of sensing the size of microbes, including *C. albicans* hyphae, to selectively release NETs [101]. An additional means of triggering NETosis is through serum IgA, with the subclass IgA2 having a significantly greater proinflammatory effect than IgA1 [102]. It is possible that an imbalance between IgA1 and IgA2 could disrupt the triggering of NET formation by intestinal fungi, potentially contributing to IBD pathogenesis [103]. NETosis is a primary mechanism through which the innate immune system can mediate fungal killing, with calprotectin acting as the predominant microbicidal peptide against *C. albicans* [93,100,104]. However, NETosis is also prone to causing collateral damage, resulting in host tissue injury and inflammation. NETs have been shown to exacerbate tissue damage in IBD [105]. TNFα stimulation induces NET formation in UC, while patients on anti-TNF medications exhibit a reduced number of NETs as well as decreased expression of NET-associated proteins in colonic tissue [105,106]. Neutrophils are clinically relevant as they form a key portion of the histological severity scores in IBD and are also a major source of calprotectin, a widely used clinical biomarker for the screening and monitoring of disease activity [107,108]. A further subgroup of myeloid cells that interact with neutrophils and function in both the innate and adaptive immune systems is the mononuclear cells.

Mononuclear phagocytes (MNP) include monocytes, macrophages, and dendritic cells. Their role within mucosal immunity is to encourage the immune tolerance of commensal organisms while eliminating pathogens in order to maintain mucosal homeostasis. The MNP phenotype and function vary between tissue compartments. In the gut, the fractalkine receptor CX3CR1 is of special interest in the context of IBD and antifungal immunity [109]. Fractalkine is a chemokine that is involved in the adhesion and chemoattraction of several types of CX3CR1+ immune cells. Fractalkine expression is upregulated in the mucosa of patients with active IBD and is modulated via IL-1β, TNF-α, and IFN-γ signalling and associated with the expansion of the CX3CR1+ MNP pool [110,111,112]. There is, however, conflicting evidence as to the effect of the fractalkine–CX3CR1 axis in different mouse models. One study reported an improvement in oxazolone-induced colitis through the administration of an anti-fractalkine monoclonal antibody with a reduction in colonic CX3CR1+ macrophages [113]. In contrast, CX3CR1 deficiency in mice resulted in a higher susceptibility to DSS colitis that was exacerbated upon *C. albicans* challenge [114]. There are likely to be fundamental differences between drug-induced CX3CR1 blockade and genetic manipulation, with the resultant imbalance in immune homeostasis causing significant variability in the pathogen control versus an exaggerated proinflammatory response.

An important feature of CX3CR1+ mononuclear phagocytes is their high expression of CLRs that trigger downstream signalling via the spleen tyrosine kinase (Syk) pathway [114,115]. CX3CR1+ MNPs induce Th17 cell differentiation and responses, induce IgG responses to *C. albicans*, and promote the activation of the inflammasome in DSS colitis [114,115]. *C. albicans* is the primary inducer of antifungal IgG in both mice and humans, which is thought to be achieved through CARD9 signalling in CX3CR1+ macrophages [116]. As previously mentioned, the *C. albicans* virulence factor candidalysin induces the activation of the NLRP3 inflammasome, leading to the cytolysis of MNPs, a potential de facto state of CX3CR1 deficiency [68]. Indeed, GWAS studies have demonstrated that CX3CR1+ polymorphisms are associated with different CD phenotypes as well as a reduced IgG ASCA positivity, highlighting their combined importance in IBD and fungal immunity [114,117]. In addition to their role within the innate immune system, MNPs form a crucial link, allowing the engagement of adaptive immune cells.

## 10. T Cells

Human antifungal immunity depends on Th17 cells, which defend mucosal barriers, and Th1 cells, which prevent fungal dissemination, with both axes being strongly linked to IBD pathology [118,119]. Th17 cells are typically proinflammatory and are characterised by the expression of the transcription factor RORγt, resulting in the production of IL-17, IL-6, and TNFα. Interestingly, patients with IL-17 primary immunodeficiencies, specifically IL-17A and IL-17F, develop characteristic *C. albicans* mucocutaneous infections, demonstrating a crucial role for this cytokine family in mediating antifungal immunity [118,120,121,122]. A seminal paper by Bacher et al. demonstrated how *C. albicans* is the major inducer of the antifungal Th17 pathway in humans, resulting in the production of IL-17A and IL-22 [123,124,125]. The observed Th17 response was more prominent in CD patients compared to healthy controls, corroborating previous research that demonstrated the excessive quantity of mucosal Th17 cells in IBD patients that correlated with the disease activity [123,126].

The links among Th17, antifungal immunity, and IBD pathogenesis are complex and challenging to model. DSS colitis in mice has been a valuable tool for enhancing our understanding of intestinal inflammation, but nonetheless, this model has major shortcomings [127]. The complexity of multi-system diseases such as CD or UC is hard to replicate, DSS colitis leads to a chemical-induced mucosal injury that does not emulate the transmural inflammation of CD, and the DSS is also likely to exert a significant impact on the microbiome. A further confounder is that *C. albicans* is not a natural commensal in rodents and therefore requires specific measures to induce replication in the murine GI tract [128]. Indeed, there are significant differences between murine and human whole-blood models infected with *C. albicans*, with murine blood being unable to reduce the fungal burden or filamentation [129]. Regardless of these challenges, important research has emerged from murine models of IBD and *C. albicans*. We know that IL-17 knockout mice develop more severe DSS colitis than wild-type animals, but the IL-17 pathway is crucial for preventing fungal mucosal overgrowth; hence, there is clearly a fine balance to be struck between the under- and overexpression of IL-17 [59,130]. Mice gavaged with *C. albicans* do not develop spontaneous inflammation; however, colonisation is promoted in the context of DSS-induced colitis, with associated increases in inflammation and immune responses directed against fungal antigens [36,131]. These data demonstrate that the Th17 axis plays a crucial role in both IBD and *C. albicans* immunity, while also highlighting gaps in our understanding. These gaps are exemplified by the failure of anti-IL17 therapy in IBD trials that is discussed later in the article.

Commensal-specific T cells regulate gut inflammation and homeostasis, promoting tolerance and epithelial barrier function [132]. They are especially important in the intestine, given that gut-resident CD4+ T cells are strongly enriched in terms of their reactivity to *C. albicans* [133]. There is a larger proportion of Th17 cells in gut-resident populations compared to circulating T cells, an effect that is more pronounced in IBD patients [133]. More recently, yeast-reactive CD4+ Th1 cells displaying cytotoxic features and IFNγ production were found to be enriched in blood and inflamed mucosa from patients with CD but not UC [134], suggesting that the fungal species recognised and the host responses induced vary significantly between disease subtypes. A recent study by Hackstein and colleagues reported on the importance of MHC-II-restricted, commensal-reactive T cells in the colons of both humans and mice. MHC-II-restricted colonic CD4+ T cells expressing TNF display above-average levels of C-type lectin CD161 and exhibit increased antimicrobial responses, including those towards *C. albicans* [135]. CD161+ CD4+ T cells were previously recognised as a Th17 cell population that is activated by IL-23 to drive intestinal inflammation [136]. These particular T cells aggravate inflammation in murine DSS colitis and are more abundant in UC patients than healthy controls, suggesting a role in IBD pathogenesis as well as antifungal immunity [135].

Tissue-resident immune cells, including various types of unconventional T cells that are strongly enriched at epithelial barrier sites, are increasingly being recognised as important and distinct components of mucosal immunity. Among these are the γδ T cells, which play important roles in antimicrobial defence but have also been implicated in IBD pathogenesis [137]. In mice, γδ T cells acquire effector functions in the thymus before populating body tissues in a characteristic pattern during early life, with distinct γ chain-bearing subsets seeding different epithelial barrier sites [138]. In humans, the major subsets are typically subclassified on the basis of δ chain usage, with Vδ1+ cells predominating in the epithelia and Vδ2+ cells primarily residing in the blood, although both subsets can populate mucosal tissues [139]. While not as numerous as conventional lymphocytes, γδ T cells modulate cytokine production by the intestinal αβ T-cell pool and can express significantly increased levels of proinflammatory cytokines in the context of IBD [140,141,142]. Similarly, the duodenal γδ T cell compartment is typically expanded in tissue-destructive coeliac disease, while various authors have reported that cell frequencies also correlate with the disease severity and therapy in IBD [140,142,143]. In mouse models, the Vδ1 subtype appears to be a major source of IL-17 that is thought to be stimulated by dendritic-cell-derived IL-23, thereby mediating protection against oropharyngeal candidiasis [144,145,146]. This is another example of an antifungal immune pathway involving IL-17 that is associated with pathological mucosal inflammation.

Aberrant type 1 immunity is heavily linked with IBD and may also increase the susceptibility to mucosal fungal infections. The importance of Th1 responses in IBD is unequivocal, with numerous studies demonstrating that mucosal inflammation is associated with increased levels of Th1 cytokines including IFN-γ, TNFα, and IL-12 [119]. Indeed, anti-TNF monoclonal antibody therapy remains one of the most effective current treatments for IBD [147]. A study focused on alterations in the mycobiota of CD patients showed a potential link between aberrant type 1 immunity and antifungal immunity [148]. *Debaryomyces hansenii* (also known as *Candida famata*) was found to colonise inflamed tissue in CD and impair mucosal healing through macrophage induction of the type 1 interferon–CCL5 axis [148]. The clinical relevance of such a relationship is evident, given that a primary treatment goal in CD is restoration of the barrier integrity and mucosal healing.

Given the number of inflammatory mechanisms involved in mucosal fungal immunity, it is perhaps unsurprising that there have been varying signals about the pathologic predominance of individual pathways. Chronic mucocutaneous candidiasis (CMC) is pathognomonic of autoimmune polyendocrine syndrome type 1 (APS-1), a monogenic deficiency in an autoimmune regulator gene. A recent paper by Break et al. reported that aberrant Th1 cell responses cause inflammation and breakdown of the mucosal integrity, leading to *C. albicans* overgrowth through CD4+ and CD8+ T cell-mediated IFNγ release [149]. Th17 responses and IL-17R/IL-22 pathways were intact and thus did not appear to limit *C. albicans* epithelial invasion, despite the presence of autoantibodies that neutralised IL-17A, IL-17F, and IL-22. However, the results of this study proved controversial, with several commentaries contesting the evidence that interferonopathy, rather than IL-17 autoantibodies, underpins CMC in APS-1 patients [150,151]. Various issues were raised concerning the study design, such as the choice of murine model and the method of determining IL-17 function, although it is possible that multiple mechanisms contribute and thus additional studies will be required to resolve these discrepancies [152,153].

## 11. Therapies

Over the last 20 years, there has been a flurry of new immunological therapies for IBD, but there are still no licensed treatments that target microbial dysregulation. Despite this, there have been promising efforts to develop faecal microbial transplantation (FMT) [154,155,156], a process that has already been proven to be effective for treating refractory *Clostridium difficile* infection [157]. Given the link between dysbiosis and IBD pathogenesis, it was a logical step for FMT to also be trialled in IBD, with randomised controlled trial (RCT) data indicating a benefit for mild to moderate UC. There is, however, significant variability in how FMT is delivered (dose, route, microbiota preparation, timing, and frequency), leading to significant challenges in assessing and comparing trials [154,158,159,160]. The most recent meta-analysis in UC has found the odds of achieving combined clinical and endoscopic remission with FMT compared to a placebo to be 4.11 (95% CI, 2.19–7.72; *p* < 0.0001) [154]. The evidence supporting FMT efficacy is weaker in CD; however, further studies are required [155].

Efforts have been made to understand the effects of FMT on the microbiomes of recipient patients, with bacterial diversity appearing to increase in individuals who respond compared with those who do not [161,162]. A fascinating study that characterised the mycobiota of FMT-treated UC patients highlighted important associations between the response and the *C. albicans* burden [163]. Elevated *C. albicans* levels in the gut prior to FMT were associated with an improved clinical response, while a decrease in the *C. albicans* burden post-FMT was linked with an improved bacterial diversity and superior clinical outcomes [163]. These findings suggest a potential role for *C. albicans* as a biomarker in predicting the FMT efficacy.

With the growing interest in *C. albicans* as a potential therapeutic target, a small number of studies have investigated the effects of oral antifungal medications in CD and UC patients. A recent pilot study was performed to assess whether 6 months of oral fluconazole could prevent post-operative recurrence in CD patients [164]. Intriguingly, fluconazole treatment was associated with reductions in biomarkers linked to recurrence. Fluconazole has been trialled as a therapy for UC on two occasions, and within the limits of the studies, there were signals that fluconazole therapy could result in biochemical and histological improvements; however, larger multi-centre RCTs are required to examine this [165,166]. However, consistent with other reports, one of these studies identified that active disease and recent steroid use were independent predictors of *C. albicans* colonisation [166]. An important consideration when administering broad spectrum antifungal drugs is that these agents target both commensal and virulent strains of *C. albicans*, which published studies rarely factor into their analyses. Despite some positive signs, oral fluconazole is unlikely to be the golden bullet for antimicrobial treatment in IBD, especially when factoring in the risks of drug-associated toxicity and fungal resistance. More targeted therapeutic approaches are needed that can preferentially deplete pathogenic over commensal strains.

*C. albicans* filamentation and adhesion to epithelial surfaces are core virulence factors and potential therapeutic targets. The induction or provision of antibodies capable of neutralising key pathogenic factors could therefore prove to be effective treatments for different subtypes of IBD. For example, the NDV-3A vaccine induces IgG and IgA responses to the adhesin Als3, which plays major roles in filamentation and binding to the epithelia. NDV-3A has already proven to be safe and effective for inducing host lymphocyte responses and reducing symptoms in a phase II RCT for recurrent vaginal candidiasis [167]. Following this promising trial, the vaccine was also shown to prevent *C. albicans*-induced damage in DSS colitis [60]. Another potential future therapy that is at a much earlier stage of development involves attempts to replicate a protective CARD9 variant. This variant is unable to recruit the binding partner TRIM62, which leads to reduced activation of NF-κB [168,169]. A small molecule has been developed that binds to CARD9 and disrupts interactions with TRIM62, resulting in the inhibition of Dectin-1-mediated, but not LPS, signalling [169]. The risks of side effects such as fungal infections have yet to be elucidated, however this is a clear example of how manipulating antifungal immune pathways could lead to effective treatments for IBD.

Given the link between IL-17 and IBD, much promise was held for the IL-17A monoclonal antibody Secukinumab. This proved to be a disappointment, as the phase II clinical trial had to be halted early after futility criteria were met [170]. The evidence that the IL-17 blockade was ineffective for CD was reinforced with a further phase II trial for brodalumab, an IL-17 receptor monoclonal antibody which, again, was terminated early given the worsening disease in the active treatment group [171]. Several explanations for the findings were suggested, including the study design and the complexity of Th17 cell biology. Given the crucial role that the IL-17 pathway plays in *Candida* immunity, J.F. Colombel and colleagues advanced the hypothesis that the failure of the IL-17 blockade in CD was due to the uncontrolled growth of intestinal *C. albicans* [172]. IL-17 inhibitors have been associated not only with opportunistic *C. albicans* infections but also with the new onset and exacerbation of IBD [173,174,175]. These adverse events highlight the intricacies of the relationship between IL-17, *C. albicans* and IBD pathogenesis.

## 12. Conclusions

*C. albicans* is an important component of IBD-associated dysbiosis. The variable expression of virulence factors between *C. albicans* strains and their subsequent impacts on the mucosal immune system have changed the way that we view their roles as commensals and pathogens. As we further our understanding of the immune mechanisms involved, there is growing evidence that pathogenic *C. albicans* strains should be future targets for IBD therapies and that the organisms themselves may be suitable predictive markers for treatments such as FMT.

## Figures and Tables

**Figure 1 jof-09-01105-f001:**
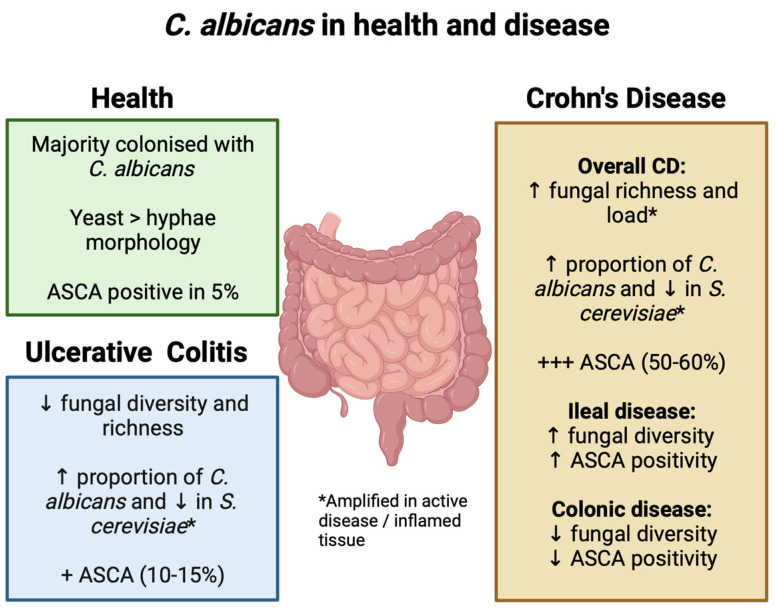
*C. albicans* in health and disease. Fungal and *C. albicans* characteristics in healthy individuals and those with ulcerative colitis and Crohn’s disease.

**Figure 2 jof-09-01105-f002:**
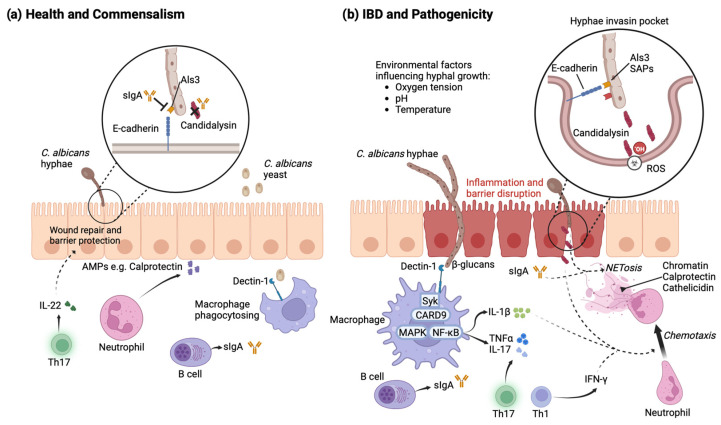
*C. albicans* mucosal immunity. Immune pathways and the effect of *C. albicans* virulence factors on (**a**) health and (**b**) IBD.

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
