# Peer review of "Mucosal Immunity to Gut Fungi in Health and Inflammatory Bowel Disease"

_jof, 2023, doi:10.3390/jof9111105_

Round 1

Reviewer 1 Report

Comments and Suggestions for Authors

This manuscript is well composed and written. It nicely discusses the relationship between mucosal immunity and gut fungi in health and inflammatory bowel disease. I would like to suggest a couple of revisions below. Please check the comments and modify your manuscript.

Major comments:

1) L249-251; I understand that sIgA is produced in Peyer’s patches and small intestinal lamina propria predominantly. However, secretory IgA is also produced by plasma cells in the colon lamina propria. In addition, this paradigm has been primarily focusing on the large intestinal inflammation so far. The authors should touch with the sIgA production in the large intestine as well.

2) L441-451; Martini and Bacher et al. recently published a paper indicating that cross-reactive T cells drive cytotoxic Th1 cell response in Crohn’s disease (https://doi.org/10.1038/s41591-023-02556-5). This article perhaps helps to improve this section in your review.

Minor comments:

1)    L50; I propose that “firmicutes” should be written as “Firmicutes”.

Author Response

This manuscript is well composed and written. It nicely discusses the relationship between mucosal immunity and gut fungi in health and inflammatory bowel disease. I would like to suggest a couple of revisions below. Please check the comments and modify your manuscript.

Many thanks to the reviewer for their positive feedback and constructive comments. We have updated the text as suggested in our revised manuscript and briefly described / cited the recent high-profile report from Martini / Bacher et al. in the corresponding section on T cell biology.

Major comments:

1) L249-251; I understand that sIgA is produced in Peyer’s patches and small intestinal lamina propria predominantly. However, secretory IgA is also produced by plasma cells in the colon lamina propria. In addition, this paradigm has been primarily focusing on the large intestinal inflammation so far. The authors should touch with the sIgA production in the large intestine as well.

Text amended in the revised version (L252-254).

2) L441-451; Martini and Bacher et al. recently published a paper indicating that cross-reactive T cells drive cytotoxic Th1 cell response in Crohn’s disease (https://doi.org/10.1038/s41591-023-02556-5). This article perhaps helps to improve this section in your review.

The recently published findings of Martini et al. have now been described and cited accordingly (L417 onwards).

Minor comments:

1)    L50; I propose that “firmicutes” should be written as “Firmicutes”.

Text amended in the revised version (L50).

Reviewer 2 Report

Comments and Suggestions for Authors

Carlson et al attempt to investigate the roles of the gut microbiome, specifically focusing on the interactions between Candida albicans, a common fungus in the human gut, and the development of inflammatory bowel disease (IBD). It highlights the importance of both bacterial dysbiosis and host immune-fungal interactions in IBD pathogenesis. The review covers the immunological impact of variations in C. albicans colonization, morphology, genetics, and proteomics in IBD, along with clinical and therapeutic implications. The text explores the association between C. albicans, ASCA (anti-Saccharomyces cerevisiae antibodies) status, and CD (Crohn's disease), noting variations in ASCA levels based on IBD type, disease location, and genetic factors. The review also discusses the shifts in the mycobiome (fungal community) in IBD compared to the general population, with variations in fungal diversity and global fungal load in different forms of IBD. The interactions between different organisms within the gut microbiome, such as the inhibitory effect of Faecalibacterium prausnitzii on C. albicans, are highlighted. The text delves into the pathobiology of C. albicans, discussing its prevalence in the human gut, genetic diversity, and the distinction between commensal and pathogenic forms. This review paper looks interesting to me. I believe the manuscript could benefit from the following major and minor comments in a satisfactory fashion, which I describe in more detail below.

Major comments:

1.     The focus on C. albicans and its role in IBD might give the impression that other factors are less significant. Although C. albicans is a major focus of research, the review should acknowledge the potential for other microorganisms to play a role in IBD.

2.     The text mentions geographical variations in mycobiome diversity but does not delve into the implications of these differences or how they might affect the generalizability of the findings. I would appreciate it if the authors could discuss more about it.

3.     The text mentions clinical and therapeutic implications but does not delve into specific recommendations or potential applications in medical practice. Adding practical insights or suggestions for future research directions could enhance the text's relevance.

Minor comments:

1.     Line 14: “host immune-fungal interactions” -> “hosts’ immune-fungal interactions”

2.     Line 36: “a sophisticated immune organ which has evolved” -> “a sophisticated immune organ that has evolved”

3.     Line 49: “The key features of bacterial dysbiosis in IBD patients is” -> “The key feature of bacterial dysbiosis in IBD patients is”

4.     Line 65: “serves as serological biomarker” -> “serves as a serological biomarker”

5.     Line 75: “ASCA has a good specificity” -> “ASCA has good specificity”

6.     Lines 86-87: “first degree relatives” -> “first-degree relatives”

7.     Line 112: “detected expansion” -> “detected the expansion”

8.     Line 114: “western cohorts” -> “Western cohorts”

9.     Line 121: “A proposed mechanism was induction” -> “A proposed mechanism was the induction”

10.  Line 250: “induced by food antigen” -> “induced by food antigens”

11.  Lines 436-437: “the Vδ1 subtype appear” -> “the Vδ1 subtype appears”

Comments on the Quality of English Language

A moderate extent of English edits are necessary.

Author Response

Thank you to the reviewer for their comments – we have addressed various typographical errors in the revised text and added new content within the constraints of article word limit. In particular, we have ensured that the updated manuscript directly acknowledges the role of non-fungal microbes in IBD (Comment #1), indicated how geographical variance in mycobiome could influence disease characteristics (Comment #2), and provided examples of how host-fungal interactions could potentially be targeted for therapy (Comment #3). Point-by-point revisions / responses are detailed below;

Major comments:

  1. The focus on albicans and its role in IBD might give the impression that other factors are less significant. Although C. albicans is a major focus of research, the review should acknowledge the potential for other microorganisms to play a role in IBD.

The introductory section describes how a diverse microbiome is necessary to regulate mucosal immunity, as well as outlining the features and putative role of bacterial dysbiosis in IBD. We have also updated the text to clarify that multiple fungal species could potentially influence IBD pathogenicity (L36-62).

  1. The text mentions geographical variations in mycobiome diversity but does not delve into the implications of these differences or how they might affect the generalizability of the findings. I would appreciate it if the authors could discuss more about it.

The updated text now specifies that geographic variance in mycobiome profiles could potentially influence IBD phenotype, severity, and therapeutic efficacy in diverse populations (L116-117). We agree that this is an interesting area for future investigation, although limited data are available at present, and article word limit precludes extensive description of these possibilities.

  1. The text mentions clinical and therapeutic implications but does not delve into specific recommendations or potential applications in medical practice. Adding practical insights or suggestions for future research directions could enhance the text's relevance.

The detailed sections on ASCA biomarkers (L63 onwards) and Therapies (L473 onwards) describe the current evidence that fungal-immune interactions can influence IBD prognosis, diagnosis, and treatment outcomes. Future interventions that might potentially target this axis are more speculative and supported by limited evidence to date. However, we agree that this area is ripe for further investigation and have updated the text with example therapeutic opportunities (i.e. induction / provision of antibodies against fungal virulence factors could prove to be effective treatments in IBD subtypes; L514-516).

Minor comments:

  1. Line 14: “host immune-fungal interactions” -> “hosts’ immune-fungal interactions”

Text amended in the revised version.

  1. Line 36: “a sophisticated immune organ which has evolved” -> “a sophisticated immune organ that has evolved”

Text amended in the revised version.

  1. Line 49: “The key features of bacterial dysbiosis in IBD patients is” -> “The key feature of bacterial dysbiosis in IBD patients is”

Text amended in the revised version.

  1. Line 65: “serves as serological biomarker” -> “serves as a serological biomarker”

Text amended in the revised version.

  1. Line 75: “ASCA has a good specificity” -> “ASCA has good specificity”

Text amended in the revised version.

  1. Lines 86-87: “first degree relatives” -> “first-degree relatives”

Text amended in the revised version.

  1. Line 112: “detected expansion” -> “detected the expansion”

Text amended in the revised version.

  1. Line 114: “western cohorts” -> “Western cohorts”

Text amended in the revised version.

  1. Line 121: “A proposed mechanism was induction” -> “A proposed mechanism was the induction”

Text amended in the revised version.

  1. Line 250: “induced by food antigen” -> “induced by food antigens”

Text amended in the revised version.

  1. Lines 436-437: “the Vδ1 subtype appear” -> “the Vδ1 subtype appears”

Text amended in the revised version.